# Whole embryonic detection of maternal microchimeric cells highlights significant differences in their numbers among individuals

Kana Fujimoto[1]*, Akira Nakajima[2], Shohei Hori[2], Naoki Irie[1]*

1 Department of Biological Sciences, Graduate School of Science, University of Tokyo, Bunkyo-ku, Tokyo, Japan, 2 Graduate School of Pharmaceutical Sciences, University of Tokyo, Bunkyo-ku, Tokyo, Japan

* irie@bs.s.u-tokyo.ac.jp (NI); kana.fujimoto@bs.s.u-tokyo.ac.jp (KF)

**Data Availability Statement:** All relevant data are within the manuscript.

**Funding:** This research project was supported in part by Grant-in-Aid for Exploratory Research (KAKENHI ID: 17K19547) and Takeda Science Foundation. This work was performed in part at One-stop Sharing Facility Center for Future Drug Discoveries in Graduate School of Pharmaceutical Sciences, the University of Tokyo.

## Abstract

During pregnancy in placental mammals, small numbers of maternal cells (maternal microchimeric cells, or MMc cells) migrate into the fetus and persist decades, or perhaps for the rest of their lives, and higher frequencies of MMc cells are reported to correlate with variety of phenomena, such as immune tolerance, tissue repair, and autoimmune diseases. While detection of these MMc cells is considered in all pregnancies, their frequency differs largely according to tissue type and disease cases, and it remains unclear whether the number of MMc cells differs significantly among embryos in normal pregnancies. Here, for the first time, we developed a whole embryonic detection method for MMc cells using transgenic mice and counted live MMc cells in each individual embryo. Using this technique, we found that the number of MMc cells was comparable in most of the analyzed embryos; however, around 500 times higher number of MMc cells was detected in one embryo at the latest stage. This result suggests that the number of MMc cells could largely differ in rare cases with unknown underlying mechanisms. Our methodology provides a basis for testing differences in the numbers of MMc cells among individual embryos and for analyzing differences in MMc cell type repertoires in future studies. These data could provide a hint toward understanding the mechanisms underlying the variety of apparently inconsistent MMc-related phenomena.

## Introduction

Generally, the body consists of its own cells that divide from a single fertilized egg (except for symbiotic microorganisms in the body). However, this does not apply to placental mammals, including humans and mice, as small numbers (around 1 in 100,000 cells in humans) of maternal cells obtained during pregnancy exist in our body [1]. This phenomenon is called maternal microchimerism (MMc), which signifies a chimera of our own cells and small numbers of genetically and immunologically non-self, maternal cells [2–4]. These maternal microchimeric

**Competing interests:** The authors have declared that no competing interests exist.

cells (MMc cells) migrate into the body via the placenta (and through milk after birth) [5] and are retained for decades after birth [1, 6, 7]. Previous studies have suggested possible beneficial aspects of MMc cells, such as establishment of immune tolerance against noninherited maternal antigens in offspring [8–11] and potential contribution to tissue repair [6, 7, 12, 13]. In addition, possible harmful effects, such as contributions to the pathogenesis or deterioration of some inflammatory diseases and congenital malformations have also been reported [14–18]. These results suggest that MMc cells could lead to opposing phenomena with unknown mechanisms. In addition, the frequency of MMc cells seems to differ significantly among individuals. Difference in the frequency of MMc cells has been reported in diseased patients and also in individuals within the same disease or control groups [6, 15]. Some patients with type I diabetes, for example, had approximately 500 MMc cells per 100,000 host cells (estimated using the amount of DNA equivalents), whereas others had undetectable levels of MMc cells. Similarly, a very high frequency of MMc cells was also found in an unaffected sibling, showing 153 MMc cells per 100,000 host cells, whereas most of the unaffected siblings showed 3.5 cells per 100,000 cells [6]. This could be due to the *post hoc* expansion of MMc cells after the onset of the disease or tissue repair; however, an alternative possibility could be that the number of MMc cells largely differed from the fetal stage or before showing any of these phenotypes. In other words, it is not known whether there are differences in the number of MMc cells among normal pregnancies. In this regard, previous approaches have a major limitation in testing this alternative possibility, as most of them focused only on MMc cells from a limited number of tissues and organs. In particular, different numbers of MMc cells reported in previous studies could merely represent a local accumulation of MMc cells in the tissue of focus, while the overall number of MMc cells in the fetus may not differ among the different individuals.

To test whether the frequency of MMc cells differs among individual embryos during normal pregnancy, we first developed a method for comprehensive isolation and counting of MMc cells from a single embryo. We succeeded in developing an effective method for detecting whole embryonic MMc cells and found that the number of MMc cells was comparable in most of the analyzed embryos; however, around 500 times higher number of MMc cells was detected in one embryo at the latest stage.

## Materials and methods

### Ethical statement

Animal care and experimental procedures were conducted in strict accordance with the guidelines approved by the School of Science, the University of Tokyo (approval ID:17–2). All efforts were made to minimize suffering. Pregnant female mice were first anesthetized by exposure to isoflurane and next euthanized by cervical dislocation.

### Mouse strains

The inbred strains, BALB/cByJJcl and C57BL/6JJcl were obtained from Clea Japan. GFP expressing mice, C57BL/6-Tg(CAG-EGFP)C14-Y01-FM131Osb was obtained from RIKEN BioResource Research Center (RBRC), which was developed by Okabe M. et al. [19] and genotyping was performed according to the instructions provided by RBRC using PCR. To obtain GFP heterozygous female mice, BALB/c female mice were mated with GFP homozygous male mice. To obtain wild type fetus with MMc cells, GFP heterozygous female mice were mated with BALB/c male mice and only fetuses without GFP genes were used for experiments (Fig 1A and 1B). Fetuses in the pregnant C57BL/6JJcl female mice were used as negative control for the MMc cell counting experiment.

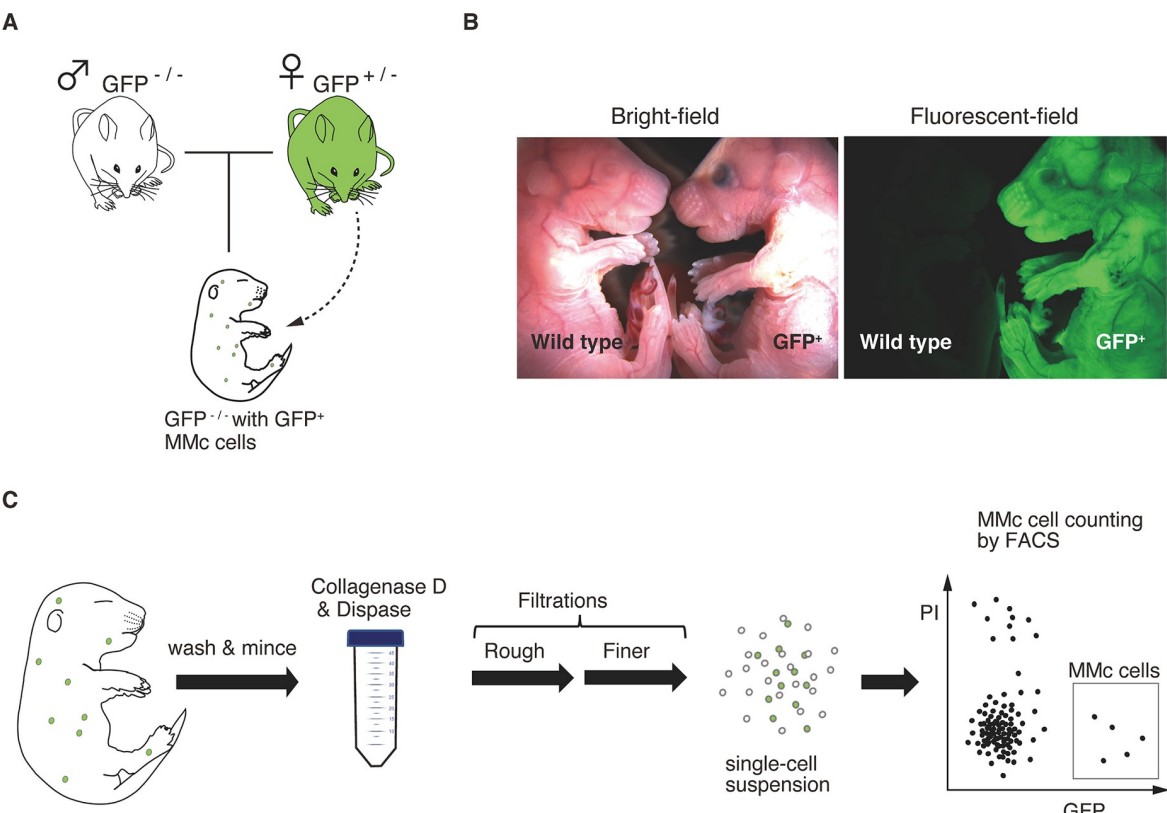

**Fig 1. Detection of MMc cells from a whole embryo.** (A) Female mice heterozygous for the GFP locus (GFP$^{+/-}$) were mated with non-GFP-carrying wild-type male mice, and offspring with no GFP gene (GFP$^{-/-}$) were analyzed for GFP$^{+/-}$ MMc cells. (B) Bright-field and fluorescent-field images showed which fetus inherited the GFP gene (right) and which did not (E18.5, litter). GFP$^{-/-}$ fetus was used for MMc cell isolation. (C) Schematic illustration of the developed method for detecting MMc cells from a whole embryo (see also Method). To avoid cross-contamination of maternal cells, an isolated embryo was washed three times in cold phosphate-buffered saline. The embryo was then minced and incubated in an enzyme cocktail, followed by two (rough and fine) filtrations to dissociate cells. Rough filtration was carried out to dissociate the solid tissues, and fine filtering was carried out to generate a single cell suspension. From this single cell suspension, live MMc cells were counted and isolated using FACS.

## Preparation of cell suspensions from whole embryo

To obtain suspensions of dissociated cells from the whole embryo, GFP$^{-}$ embryos (identified by the absence of fluorescence using GFP excitation flashlight while embryos are in amnion) were carefully dissected from their sacrificed mother to avoid cross-contamination of fetal and maternal cells. In brief, pregnant females were first sacrificed and uteruses were cut out to obtain embryos with amnion, and placed in cold phosphate-buffered saline. After transferring the embryo to a new cold phosphate-buffered saline, amnions were removed from embryos, followed by a cut of the umbilical cord. Embryos were then washed with cold phosphate-buffered saline 3 times by picking the fetal side remaining umbilical cord. To obtain dissociated cells, each embryo was first minced using a scissors and further dissociated by incubating in solution with 1 mg/ml collagenase D (Roche), 2.4 mg/ml Dispase (gibco), 100 U/ml DNase I (Roche) in HBSS(+)+3%FBS for 20 min at 37˚C. This enzyme cocktail was made by referring to Mass E. et al. [20]. The minced tissues were then filtered by squeezing into 100-μm cell strainers using the rubber part of the syringe (Rough Filtration). The filtered samples were centrifuged at 1,300 r.p.m. for 7 min at 4˚C followed by Ammonium-Chloride-Potassium (ACK) lysis buffer treatment for 6 min at r.t. After removing erythrocytes, cold (on-ice) FACS buffer

(HBSS(-), 0.5% BSA, 2mM EDTA) was added to the cell suspension and centrifuged again with the same condition (1,300 r.p.m. for 7 min at 4˚C). To generate single-celled suspension, the cell suspension was filtered again with 70-μm cell strainers (Finer Filtration). The number of cells in the cell suspension was counted and adjusted to $1.0 \times 10^7$ cells/ml concentration for the sorting process.

To create positive control for MMc cells, blood of the sacrificed mother was collected. For these maternal cells, erythrocyte removal was conducted twice because the amount of red blood cells are much more abundant than fetus samples. First round removal was performed with the same method done for the fetus samples, and in the second round with ACK treatment for 3 min. The embryos from the same developmental stage as sample embryos were used as negative control. For the control data for validating cell survival rate, cell suspension obtained from adult spleen was generated.

## Rough and finer filtrations

Cell strainers (100-μm and 70-μm, Falcon) were used for both rough and finer filtrations. Whether or not the aggregation after the filtration mainly consists of dead cells were evaluated by trypan blue staining (100 μl trypan blue to the 5 ml cell suspension). Ratio of live cells were further analyzed with FACS (BD AriaⅢu) with PI (Dojindo) staining.

## MMc cell counting and observation

The cells ($> 10,000,000$ cells, which roughly corresponds to one-fourth of total cell suspension volume) obtained from a single fetus were stained with Propidium Iodide (PI) solution, and were loaded into a fluorescence-activated cell sorting (FACS) machine (BD AriaⅢu). After removing doublet cells and separating the live cell and dead cell populations by PI marker, the number of GFP positive live cells (GFP$^+$ PI$^-$ cells) was counted as the number of MMc cells (Fig 1C). The frequency of MMc cells was calculated and normalized to the number of MMc cells per 10,000,000 fetal (GFP$^-$ PI$^-$) cells. Sorted MMc cells were transferred into a glass-bottom dish and observed by fluorescence confocal microscopy (Zeiss LSM710) to check positive signal for the GFP and negative signal for the PI.

## Results

### Development of whole embryonic MMc cell detection protocol

To identify maternal cells from the whole embryo, we took advantage of the green fluorescent protein (GFP) mouse strain. In brief, by crossing female mice heterozygous for the GFP locus (GFP$^{+/-}$) with wild-type male mice, we obtained wild-type fetuses (GFP$^{-/-}$) with GFP-carrying MMc cells (Fig 1A and 1B). Wild-type (GFP$^{-/-}$) embryos harboring GFP$^{+/-}$ MMc cells were then carefully dissected from their mother by repeated wash processes (three times with phosphate-buffered saline) and subjected to protease (trypsin) digestion, followed by two rounds (rough and fine) of physical filtration using cell strainers to dissociate cells. Finally, maternal cells were detected using fluorescence-activated cell sorting (FACS) on this cell suspension (Fig 1C).

### The method performs well for developmental stages E12.5 to E15.5

We then analyzed which developmental stage of the mouse would be best suited for efficient isolation of MMc cells. MMc cells have been detected as early as E12.5–E13.5 in mice [21, 22], and bi-directional trafficking of MMc cells is often considered to increase throughout gestation, peaking at parturition [23]. Thus, we decided to target the developmental stages from

E12.5 to E18.5. While later developmental stages are expected to have more MMc cells [23], dissociating cells would be more difficult because of the solid tissues enriched with extracellular matrices. As expected of the rough filtration process, minced samples from later developmental stages (E16.5 and E18.5) tended to retain more debris (presumably bones and tendons) after the filtration process compared with earlier stages (Fig 2A). In contrast, the samples from earlier stages (E12.5–E15.5) were digested relatively easily by the filtering process. For the finer filtration process, although samples from all stages retained some aggregates, a relatively larger amount remained in E16.5 and E18.5 samples. After staining the pre-filtered sample with trypan blue, the filtered cells were unstained, while aggregates were stained blue (Fig 2C), suggesting that they mainly consisted of dead cells or dead cells with extracellular matrices. The filtered cells were then analyzed for the ratio of live cells using PI markers and FACS (Fig 2D). Our results indicated that the ratio of live cells was comparable with that in control samples from the spleen, suggesting that a fair number of live cells can be obtained following our dissociation protocol. Considering that the aggregates remained after the two filtering processes for the late-stage samples (E16.5 and E18.5), it would be reasonable to conclude that our method is best suited for stages E12.5–E15.5, obtaining the maximum number of live cells.

## Possible differences in frequency of MMc cells among individual fetuses

After obtaining the cell suspension from a single mouse embryo using the filtering method described above, we further analyzed and isolated MMc cells by FACS. We first removed dead and doublet cells using PI staining and gating. As shown in Fig 3A, noise signals and/or debris were removed with gate P1, and then doublet cells were removed with gates P2 and P3. P4 gate setting for the GFP$^+$ PI$^-$ (live MMc cells) was performed by checking the signal intensity of positive control cells (cells from maternal blood). Around $1 \times 10^7$ cells were loaded into the FACS, and the GFP$^+$ PI$^-$ cells were counted. Finally, the frequency of MMc cells was calculated as the number of MMc cells over $10^7$ PI$^-$ cells (Fig 3B). Although some false positive signals were detected in the samples from E14.5 negative controls, no GFP$^+$ cells were detected in the negative control samples (cells prepared from a wild-type embryo with a wild-type mother) for the other stages. In contrast, many GFP$^{-/-}$ embryos with GFP$^{+/-}$ mothers contained GFP$^+$ cells. GFP signals of these potential maternal cells in the GFP$^{-/-}$ embryos were also confirmed using microscopy (Fig 3C). These results indicate that maternal cells were successfully detected using our method in a single embryo. Meanwhile, GFP$^+$ cells were not detected in some of the GFP$^{-/-}$ embryos. Given that all embryos are considered to have maternal cells [22], the results suggest that our method is not sensitive enough to detect MMc cells in some of the embryos. Additionally, given the false positive signals in the negative control samples, it is also possible that some false positive signals were contained in the tested samples. Nevertheless, the frequencies of GFP$^+$ signals in the tested samples were higher than those in the negative control samples, and one sample from stage E18.5, showed as high as 1800 cells over $10^7$ cells. Together, these results suggest that our method detects MMc cells to a higher level than negative control samples. It is also worth noting that the frequency of maternal cells may differ largely in some rare cases, as for the stage E18.5 embryo.

## Discussion

In this study, we developed a method for counting and isolating MMc cells from whole mouse embryos through detecting GFP$^+$ cells to test if the number of maternal cells differed among individual embryos (Figs 1 and 2). Although we could not completely exclude the false positive signals from contaminated maternal cells during the experimental procedure, the overall signals were higher in the test samples than in the negative control samples (Fig 3B). Based on

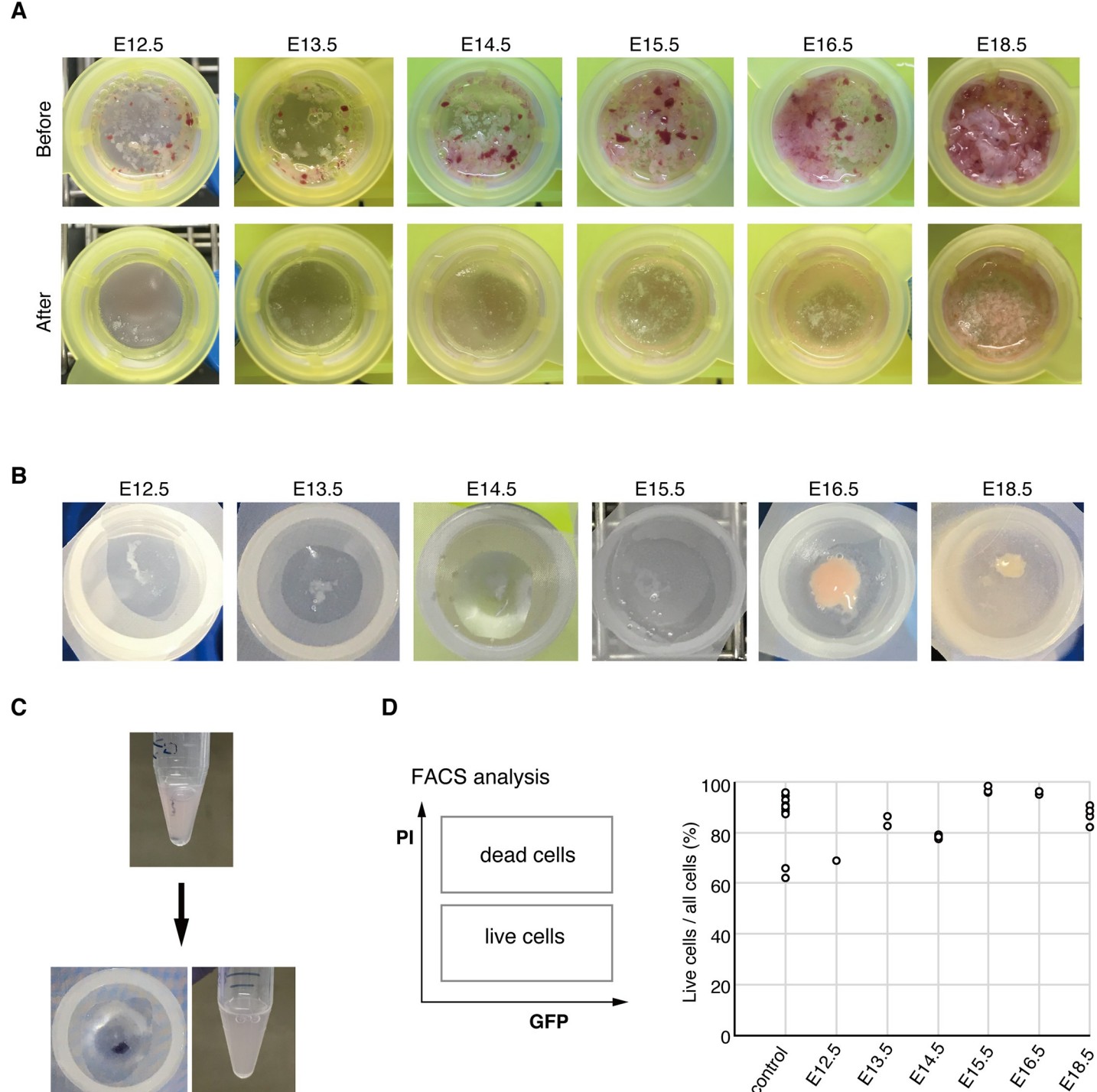

**Fig 2. Dissociation of cells by filtration and survival rate of isolated cells.** (A) Minced samples before and after the rough filtration are shown for each developmental stage. A 100-μm cell strainer was used for the rough filtration. Note that more aggregates remained in samples from the later stages E16.5 and E18.5. (B) Samples after fine filtration with the 70-μm cell strainer. (C) Rough filtered samples were stained with trypan blue, and finer filtration was performed to estimate whether the aggregates were mainly composed of dead cells. (D) Live cell rate in each developmental stage. Calculation of live cell rate (%) in the cell suspensions was evaluated using PI marker (dead cell marker) and FACS. The control sample originated from an adult spleen. Each circle represents the ratio of live cells in a single sample. [control: n = 11, E12.5: n = 1 (68.8%), E13.5: n = 2 (median 84.3%), E14.5: n = 5 (median 78.4%), E15.5: n = 4 (median 97.0%), E16.5: n = 2 (median 95.4%), E18.5: n = 4 (median 87.0%)].

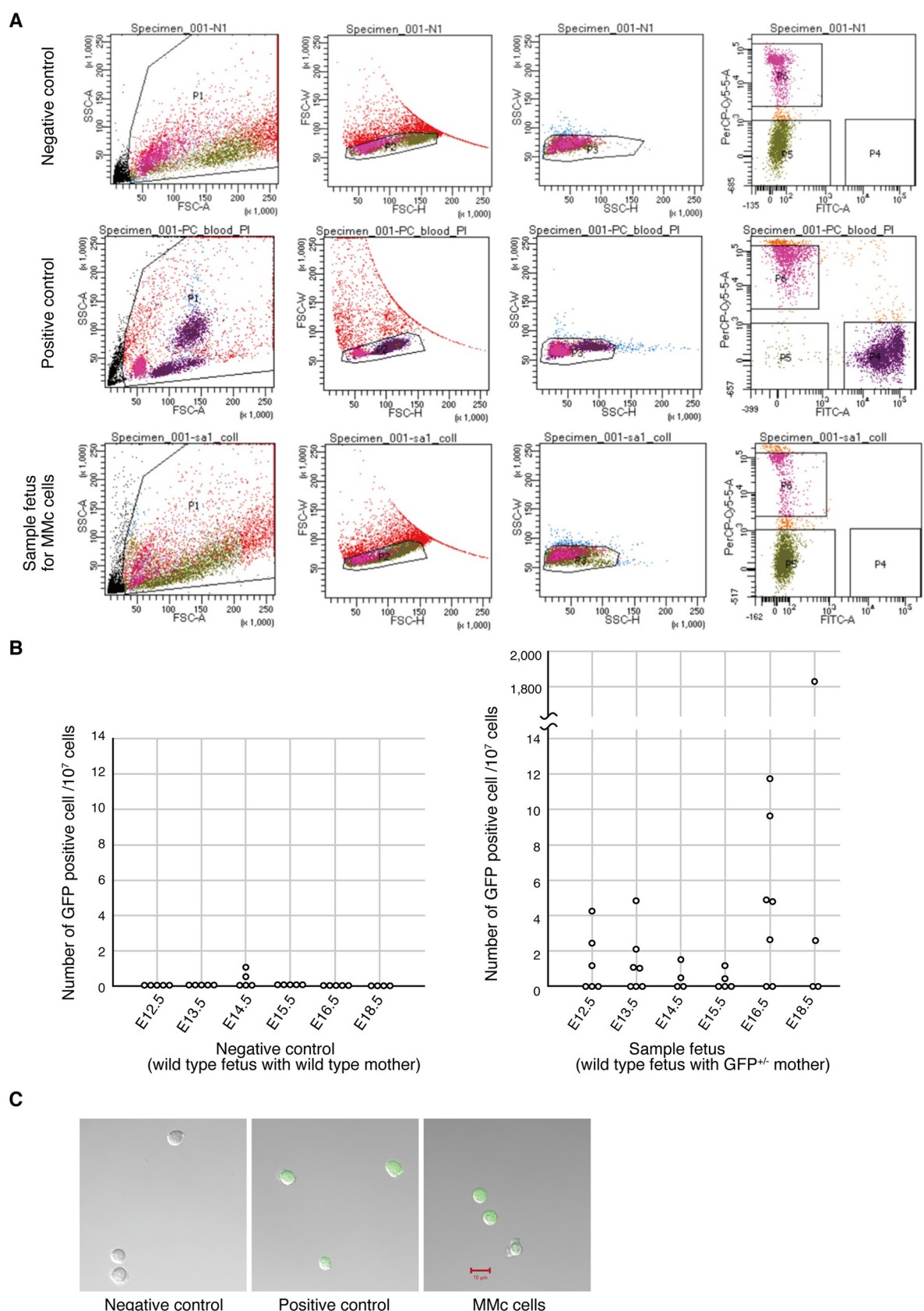

**Fig 3. FACS detection of MMc cells from individual embryos.** (A) Gating conditions for counting MMc cells. After the rough and finer filtration of the minced embryo, isolated cells from a single GFP$^{-/-}$ embryo were analyzed using FACS. Mother blood cells were used as positive control, and cells from a wild-type fetus with a wild-type mother were used as negative control. Noise and/or debris were removed with gate P1, and then doublet cells were removed with gate P2 and P3. Gate P4 (GFP$^+$ PI$^-$) was defined using positive control cells and applied to sort the GFP$^+$, potential MMc cells. (B) For each sample (individual embryo), the numbers of sorted MMc cells were counted as GFP$^+$ cells, and the ratios over the 10$^7$ sorted cells were calculated. Some GFP$^+$ cells were detected for the negative control samples (wild-type embryo with wild-type mother). Sample numbers are as follows. control: E12.5: n = 5, E13.5: n = 5, E14.5: n = 5, E15.5: n = 5, E16.5: n = 5, E18.5: n = 4, sample: E12.5: n = 6, E13.5: n = 7, E14.5: n = 4, E15.5: n = 5, E16.5: n = 7, E18.5: n = 4. See also Table 1 for more detail. (C) GFP signals of sorted cells were further analyzed using fluorescent confocal microscopy (Zeiss LSM710). Together with the sample cells, negative (wild-type cells sorted with gate P5) and positive control (mother blood cells sorted with gate P4) cells were also analyzed using microscopy.

these results, we found that majority of the embryos showed a comparable number of MMc cells. Meanwhile, an unexpected finding was that one of the embryos at the latest stage (E18.5) showed 1,816 cells/10$^7$ cells, which corresponds to a frequency approximately 500 times higher than the average of the other detected embryos (around 3.7 cells/ 10$^7$ sorted cells for those detected with GFP$^+$ cells but without the one with 1816 cells. See Fig 3B and Table 1). We suspected a possible contamination of mother blood cells during the sample preparation process for this sample; however, considering that the sample preparation protocol comprises multiple careful washing steps and that no GFP$^+$ cells were detected in the vast majority of the negative controls. To add, another possibility was that the GFP$^+$ cells identified here could be from GFP$^{+/-}$ siblings in the same liter, however, considering that the litter with the sample had relatively lower ratio of GFP$^{+/-}$ siblings compared to those of the other stages (the ratio of GFP$^{+/-}$: GFP$^{-/-}$ embryos of the E18.5 was 0.23:1 in the sample litter, where it was 1:1.2 on average in the litters of the other stages. See also Table 1), and we also avoided using embryos with conjugated placenta, the probability of this sibling origin scenario appears to be less likely. Even a minor possibility would be that the embryo was a tetragametic chimera, leading to the high frequency of GFP$^+$ cells, however, at least we could not find any sign of abnormality, including body size, morphology, and patchy signals of GFP. Thus, it would be fair to conclude that most of the cells detected here were MMc cells or at least immunologically non-self cells. Finally, considering that later stages are known to show higher frequency of MMc cells [23], it is intriguing that the sample with highly frequent MMc cells was detected for the latest embryonic stage. Consistently, while no statistically significant correlation was obtained between the number of MMc cells and developmental stages, however, we still found a weak tendency even without stage 18.5 (Spearman's correlation coefficiency 0.3 for stages from 12.5 to 16.5). Together, these results suggest that while most of the embryos have a comparable number of MMc cells, a much higher count can be detected in some rare cases. Given this finding, it is intriguing to know if these high-MMc embryos are associated with a variety of MMc-related phenomena that appear to be contradictory to each other [6–18]. With this respect, it is tempting to know if the ratio of MMc$^+$ detected embryos and those undetected differ among the litters (e.g., the litters at stages E13.5 and E16.5 show significant variation in the proportion of MMc+ detected embryos, see also Table 1). Furthermore, our method provides a basis for comprehensively analyzing the differences in cell type repertoires of MMc cells among individual embryos, such as using FACS or single cell RNA-seq technology, which could be applied in future studies. Future studies that combine these technologies would provide a quantitative approach for testing additional hypotheses, such as whether the cell type repertoire of MMc cells differs among individual embryos, which could explain seemingly contradictory phenomena related to microchimerism. Finally, major caveats of our methodology would be that 7–8 hours are needed to process embryos to obtain single celled suspension, and is not best suited to the mouse embryos from the latest stages (Fig 2A and 2B). This, together with automatically

**Table 1. Detailed information of GFP+ and GFP- fetal samples in Fig 3B.**

| stages | litters | # of total embryos | # of GFP+ and GFP- embryos | GFP- fetus | # of detected GFP + cells | # of total sorted cells (processed events count) | Frequency of GFP+ cells (/10^7) |
|---|---|---|---|---|---|---|---|
| E12.5 | Litter A | 11 | 7 : 4 | A_a | 0 | 14845653 | 0 |
| | | | | A_b | 0 | 15583576 | 0 |
| | | | | A_c | 3 | 12591640 | 2 |
| | | | | A_d | 5 | 11957752 | 4 |
| | Litter B | 4 | 2 : 2 | B_a | 0 | 15130672 | 0 |
| | | | | B_b | 2 | 17418976 | 1 |
| E13.5 | Litter C | 10 | 6 : 4 | C_a | 4 | 8363224 | 5 |
| | | | | C_b | 0 | 19424512 | 0 |
| | | | | C_c | 0 | 13583552 | 0 |
| | | | | C_d | 0 | 11746550 | 0 |
| | Litter D | 12 | 4 : 8 | D_a | 1 | 9638845 | 1 |
| | | | | D_b | 2 | 20525153 | 1 |
| | | | | D_c | 3 | 14678961 | 2 |
| E14.5 | Litter E | 9 | 4 : 5 | E_a | 2 | 13667524 | 1 |
| | | | | E_b | 1 | 21182691 | 0 |
| | Litter F | 10 | 8 : 2 | F_a | 0 | 17686309 | 0 |
| | | | | F_b | 0 | 15668245 | 0 |
| E15.5 | Litter G | 11 | 4 : 7 | G_a | 2 | 17827792 | 1 |
| | | | | G_b | 1 | 23348680 | 0 |
| | | | | G_c | 0 | 18040216 | 0 |
| | | | | G_d | 0 | 33750160 | 0 |
| | | | | G_e | 0 | 20301805 | 0 |
| E16.5 | Litter H | 14 | 5 : 9 | H_a | 8 | 8409021 | 10 |
| | | | | H_b | 5 | 19457597 | 3 |
| | | | | H_c | 10 | 20648028 | 5 |
| | | | | H_d | 10 | 21205760 | 5 |
| | | | | H_e | 19 | 16388294 | 12 |
| | Litter I | 2 | 0 : 2 | I_a | 0 | 16528251 | 0 |
| | | | | I_b | 0 | 21958902 | 0 |
| E18.5 | Litter J | 9 | 2 : 7 | J_a | 5 | 19551458 | 3 |
| | | | | J_b | 0 | 24253026 | 0 |
| | | | | J_c | 0 | 14541005 | 0 |
| | | | | J_d | 2686 | 14786949 | 1816 |

removed cells in FACS as an electronic aborts (around 1 in 100 counts in our experiments) may explain the reason why we could not detect GFP+ cells in some of the embryos. Furthermore, while our experimental design cannot exclude the possibility of detecting GFP+ cells from GFP+/- siblings in the same litter, this could be solved by crossing mother having GFP+/RFP+ on the same locus, with non fluorescent wild type male. In this design, GFP/RFP double positive cells in either GFP+/- or RFP+/- fetus should always be the cells of maternal origin.

## Acknowledgments

This work was performed in part at One-stop Sharing Facility Center for Future Drug Discoveries in Graduate School of Pharmaceutical Sciences, the University of Tokyo. We would like to thank Editage (www.editage.com) for help editing English. We acknowledge one of the reviewers for providing a specific idea for discriminating maternal cells from cells of siblings.

## Author Contributions

**Conceptualization:** Kana Fujimoto, Naoki Irie.

**Funding acquisition:** Naoki Irie.

**Investigation:** Kana Fujimoto, Akira Nakajima, Shohei Hori, Naoki Irie.

**Methodology:** Kana Fujimoto, Naoki Irie.

**Project administration:** Kana Fujimoto, Naoki Irie.

**Supervision:** Akira Nakajima, Shohei Hori.

**Validation:** Kana Fujimoto, Naoki Irie.

**Writing – original draft:** Kana Fujimoto, Naoki Irie.

**Writing – review & editing:** Kana Fujimoto, Shohei Hori, Naoki Irie.

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
