## [Decision Letter · Decision Letter 0]

31 Aug 2021

PONE-D-21-14310

Whole embryonic detection of maternal microchimeric cells highlights significant differences in their numbers among individuals

PLOS ONE

Dear Dr.Naoki Irie ,

Thank you for submitting your manuscript to PLOS ONE. After careful consideration, we feel that it has merit but does not fully meet PLOS ONE’s publication criteria as it currently stands. Therefore, we invite you to submit a revised version of the manuscript that addresses the points raised during the review process.

Please address all issues raised by both reviewers, and in particular, authors must demonstrate that microchimeric cells are all of maternal origin as GFP+ cells can also come from another fetus of the same litter carrying the maternal GFP.

We look forward to receiving your revised manuscript.

Kind regards,

Colette Kanellopoulos-Langevin, Ph.D

Academic Editor

PLOS ONE

1. Please ensure that your manuscript meets PLOS ONE's style requirements, including those for file naming. The PLOS ONE style templates can be found at https://journals.plos.org/plosone/s/file?id=wjVg/PLOSOne_formatting_sample_main_body.pdf and https://journals.plos.org/plosone/s/file?id=ba62/PLOSOne_formatting_sample_title_authors_affiliations.pdf.

“This research project was supported in part by Grant-in-Aid for Exploratory Research (KAKENHI ID: 17K19547) and Takeda Science Foundation. This work was performed in part at One-stop Sharing Facility Center for Future Drug Discoveries in Graduate School of Pharmaceutical Sciences, the University of Tokyo. We would like to thank Editage (www.editage.com) for help editing English.”

“This research project was supported in part by Grant-in-Aid for Exploratory Research (KAKENHI ID: 17K19547) and Takeda Science Foundation. This work was performed in part at One-stop Sharing Facility Center for Future Drug Discoveries in Graduate School of Pharmaceutical Sciences, the University of Tokyo.”

“NO authors have competing interests”

5. We note that Figure 1b in your submission contain copyrighted images. All PLOS content is published under the Creative Commons Attribution License (CC BY 4.0), which means that the manuscript, images, and Supporting Information files will be freely available online, and any third party is permitted to access, download, copy, distribute, and use these materials in any way, even commercially, with proper attribution. For more information, see our copyright guidelines: http://journals.plos.org/plosone/s/licenses-and-copyright.

a. You may seek permission from the original copyright holder of Figure 1b to publish the content specifically under the CC BY 4.0 license.

Additional Editor Comments (if provided):

Reviewers' comments:

Reviewer's Responses to Questions

**Comments to the Author**

1. Is the manuscript technically sound, and do the data support the conclusions?

Reviewer #1: Partly

Reviewer #2: Partly

2. Has the statistical analysis been performed appropriately and rigorously? 

Reviewer #1: N/A

Reviewer #2: I Don't Know

3. Have the authors made all data underlying the findings in their manuscript fully available?

Reviewer #1: No

Reviewer #2: No

4. Is the manuscript presented in an intelligible fashion and written in standard English?

Reviewer #1: Yes

Reviewer #2: Yes

5. Review Comments to the Author

Reviewer #1: The authors present a whole embryonic detection method for maternal microchimeric (MMc) cells using transgenic mice to test whether the number of maternal cells differs from one embryo to another. They justify their method of detection on the whole embryo instead of testing separated organs and tissues, as they hypothesize that if MMc does not differ in number from one embryo to another the overall number of MMc cells in the fetus may not differ among the different individuals.

And they further explain that different numbers of MMc cells have been reported among various tissues in previous studies and could reflect a particular accumulation in a specific tissue. They therefore want to overcome this by testing the entire embryo.

For this purpose, authors have crossed female mice heterozygous for the GFP locus (GFP+/-, from F1 BALB/c x C57BL/6-GFP) with non-GFP BALBc male mice, and offspring with no GFP gene (GFP-/-) were analyzed for GFP+/- MMc cells.

The major inconvenient with this system is that fetuses GFP-/- may have received microchimeric cells GFP+/- from other fetuses from the same litter. Therefore authors cannot assert that GFP-/+ cells are maternal.

Moreover they observed that “MMc” differed in frequency among individual fetuses.

The authors need to provide information about the other fetuses from the same litter.

Were there more GFP + fetuses in the litter where one of them was very positive?

How many fetuses from the same litter were analyzed?

It is very difficult to discuss these results until we have this information.

To absolutely demonstrate that maternal cells are maternal cells and not cells from other fetuses I would suggest to construct another model of MMc detection. Considering that this is technically feasible, one way out might be to realize the same experiences with mothers being heterozygous for Green and Red fluorescence, thus MMc would be GFP+/- RFP+/- among either heterozygous GFP+/- RFP -/- embryos or among heterozygous GFP-/- RFP +/- embryos.

Finally, a second important point: although this is a count of GFP + cells on a whole embryo, since the detection technique is done by FACS, it is a pity that other cell markers are not added to determine the phenotype. A multiparameter analysis would give additional indications as to the type of Mc cells.

The study, nevertheless very interesting, must be reconsidered by the authors.

Reviewer #2: In the manuscript “Whole embryonic detection of maternal microchimeric cells highlights significant differences in their numbers among individuals” the authors introduce a technique to examine the quantity of allogeneic cells in a murine model. Utilizing a breeding strategy to produce GFP- embryos in a GFP+ mother, the authors were able to utilize GFP+ cell presence in the embryos as an indicator of maternal microchimerism. To measure the quantity of GFP+ cells in the embryos they implemented a fluorescence-activated cell sorting strategy to filter cells on PI staining (for live cell detection) and GFP (for allogeneic cell detection). The presence of GFP+ cells, presumably from the mother, were identified in several of the embryos with one presenting with a notably larger proportion of GFP+ cells. This technique provides a useful tool for the analysis of maternal microchimerism in embryo development and may help to answer some of the enigmas of maternal microchimerism. This is an interesting report on a technique that can aid researchers in the understanding of fetomaternal chimerism and mechanisms in utero. I found the methods to be generally well described and took particular interest in the study design. I also have some comments for the authors to consider for further improvements in clarity and dissemination of their findings:

Major comments:

1. The sample size of the study is not clear from the text or figures. Samples sizes listed in lines 196-198 suggest a smaller number of samples compared to figure 3B. It would be helpful to clearly indicate how many litters and embryos were studied at each gestational age for the experiments.

2. The results section should include either a detailed section or table containing descriptive statistics for each gestational age group. As the study focus is in the prevalence and proportion of GFP cells at different gestational ages, I would suggest this to include data on GFP cell proportions, measure of variation, proportion of embryos with detectable GFP cells, sample size, etc.

3. Line 91: The text states that “GFP+ embryos (identified by GFP excitation flashlight while embryos are in amnion) were carefully dissected from their sacrificial mother to avoid cross contamination of fetal and maternal cells.” As I read the rest of the study it refers to the study of the GFP- embryos for the presence of GFP+ cells. Is this a mistake or were the GFP+ embryos removed to avoid contamination of the GFP- embryos? If this is correct, perhaps this could be clarified by including in the schematic of Figure 1C.

4. Throughout the manuscript there is a continuous assumption that the source of the GFP+ cells is the mother, however there are other hypotheses described in the literature that could explain the source of microchimerism in utero. Assuming 50% of embryos in the litter are heterozygous (GFP+/-) it is difficult to determine the precise source of the cells. Anastomoses of placental vasculature has been previously demonstrated to produce twin-twin transfusion and could be another potential source of allogenic cells in utero. Were there any other measures used to confirm that the cells are of maternal origin? Is there any information or analyses that can be included about the heterozygous GFP+/- prevalence of each litter?

5. Lines 214-217: Could you elaborate on the possible causes of the seemingly low number of GFP+ cells measured in the embryos? This could perhaps be added as part of a broader discussion of limitations.

6. Lines 249-255: It is stated that the large proportion of GFP cells identified in one sample is a product of the mother, but it should be considered that this may be a result of tetragametic chimerism (the fusion of two embryos in utero) or possible transfusion from another embryo.

Minor comments

1. Lines 17-19: In the abstract it is stated that “maternal cells…migrate into the fetus and persist for the rest of their lives”, this appears to be an overstatement. While possible, there does not seem to be sufficient evidence to support this claim. It may be best to rephrase this into a phrase similar to “may persist for several decades” (see line 42).

2. Line 127: It is not clear if then entire material from each embryo was loaded onto the FACS machine or only a portion.

3. In lines 246-249 it is stated that this study identified a comparable number of MMc cells of previous studies reviewed by Kinder et al. However, on review the review article by Kinder et al. does not appear to summarize maternal microchimerism levels (although it does for fetal microchimerism). In this case it would be better to cite the specific studies that produced comparable results.

6. PLOS authors have the option to publish the peer review history of their article (what does this mean?). If published, this will include your full peer review and any attached files.

Reviewer #1: No

Reviewer #2: No

---

## [Author Response · Author response to Decision Letter 0]

17 Sep 2021

Response to Reviewers

5. Review Comments to the Author

Reviewer #1: The authors present a whole embryonic detection method for maternal microchimeric (MMc) cells using transgenic mice to test whether the number of maternal cells differs from one embryo to another. They justify their method of detection on the whole embryo instead of testing separated organs and tissues, as they hypothesize that if MMc does not differ in number from one embryo to another the overall number of MMc cells in the fetus may not differ among the different individuals.

And they further explain that different numbers of MMc cells have been reported among various tissues in previous studies and could reflect a particular accumulation in a specific tissue. They therefore want to overcome this by testing the entire embryo.

For this purpose, authors have crossed female mice heterozygous for the GFP locus (GFP+/-, from F1 BALB/c x C57BL/6-GFP) with non-GFP BALBc male mice, and offspring with no GFP gene (GFP-/-) were analyzed for GFP+/- MMc cells.

The major inconvenient with this system is that fetuses GFP-/- may have received microchimeric cells GFP+/- from other fetuses from the same litter. Therefore authors cannot assert that GFP-/+ cells are maternal.

>Thank you for your constructive comment. We totally agree that we cannot fully exclude the possibility that GFP+ cells detected in embryos contain those of siblings in the same litter. We have thus made the following changes to our manuscript. 

<Discussion> l. 262 ~ 272 in ‘Manuscript’

To add, another possibility was that the GFP+ cells identified here could be from GFP+/- siblings in the same liter, however, considering that the litter with the sample had relatively lower ratio of GFP+/- siblings compared to those of the other stages (the ratio of GFP+/- : GFP-/- embryos of the E18.5 was 0.23:1 in the sample litter, where it was 1:1.2 on average in the litters of the other stages. See also Table 1), and we also avoided using embryos with conjugated placenta, the probability of this sibling origin scenario appears to be less likely. Even a minor possibility would be that the embryo was a tetragametic chimera, leading to the high frequency of GFP+ cells, however, at least we could not find any sign of abnormality, including body size, morphology, and patchy signals of GFP. Thus, it would be fair to conclude that most of the cells detected here were MMc cells, or at least immunologically non-self cells.

In addition, we have added the following sentence according to the other comment.

<Discussion> l. 287 ~ 291 in ‘Manuscript’ 

Furthermore, while our experimental design cannot exclude the possibility of detecting GFP+ cells from GFP+/- siblings in the same litter, this could be solved by crossing mother having GFP+/RFP+ on the same locus, with non fluorescent wild type male. In this design, GFP/RFP double positive cells in either GFP+/- or RFP+/- fetus should always be the cells of maternal origin. 

Since the above idea is not our original, we have clarified that point in the acknowledgements section.

<Acknowledgement> l. 299 ~ 301 in ‘Manuscript’ 

We acknowledge one of the reviewers for providing a specific idea for discriminating maternal cells from cells of siblings.

Moreover they observed that “MMc” differed in frequency among individual fetuses. The authors need to provide information about the other fetuses from the same litter. Were there more GFP + fetuses in the litter where one of them was very positive? How many fetuses from the same litter were analyzed? It is very difficult to discuss these results until we have this information.

> We fully agree with the reviewer, and we have thus added the detailed information in the manuscript as Table 1. According to the table, the E18.5 embryo with the very high frequency of GFP+ cells had comparable, or even lower ratios of GFP+/- siblings (average ratio of GFP+/- : GFP-/- was 1:1.2, while the sample litter at E18.5 was 0.23:1). We have added the following sentences in the discussion. 

<Discussion> l. 262 ~ 272 in ‘Manuscript’

To add, another possibility was that the GFP+ cells identified here could be from GFP+/- siblings in the same liter, however, considering that the litter with the sample had relatively lower ratio of GFP+/- siblings compared to those of the other stages (the ratio of GFP+/- : GFP-/- embryos of the E18.5 was 0.23:1 in the sample litter, where it was 1:1.2 on average in the litters of the other stages. See also Table 1), and we also avoided using embryos with conjugated placenta, the probability of this sibling origin scenario appears to be less likely. Even a minor possibility would be that the embryo was a tetragametic chimera, leading to the high frequency of GFP+ cells, however, at least we could not find any sign of abnormality, including body size, morphology, and patchy signals of GFP. Thus, it would be fair to conclude that most of the cells detected here were MMc cells, or at least immunologically non-self cells.

To absolutely demonstrate that maternal cells are maternal cells and not cells from other fetuses I would suggest to construct another model of MMc detection. Considering that this is technically feasible, one way out might be to realize the same experiences with mothers being heterozygous for Green and Red fluorescence, thus MMc would be GFP+/- RFP+/- among either heterozygous GFP+/- RFP -/- embryos or among heterozygous GFP-/- RFP +/- embryos.

> We sincerely appreciate the proposal of this sharp experimental design. Although we do not have sufficient time to perform this additional experiment in the limited time under persisting COVID-19 situation, we have added this idea to the final discussion part of our manuscript as a future study. We also acknowledged the reviewer for this part to clarify his/her inspiring comment.

<Discussion> l. 287 ~ 291 in ‘Manuscript’

Furthermore, while our experimental design cannot exclude the possibility of detecting GFP+ cells from GFP+/- siblings, this could be solved by crossing mother having GFP+/RFP+ on the same locus, with non fluorescent wild type male. In this design, GFP/RFP double positive cells in either GFP+/- or RFP+/- fetus should always be the cells of maternal origin. 

> Since the above idea is not our original, we have clarified that point in the acknowledgements section.

<Acknowledgement> l. 299 ~ 301 in ‘Manuscript’ 

We acknowledge one of the reviewers for providing a specific idea for discriminating maternal cells from cells of siblings.

Finally, a second important point: although this is a count of GFP + cells on a whole embryo, since the detection technique is done by FACS, it is a pity that other cell markers are not added to determine the phenotype. A multiparameter analysis would give additional indications as to the type of Mc cells. The study, nevertheless very interesting, must be reconsidered by the authors.

> We thank the reviewer for this inspiring and constructive comment, as this is exactly what we originally wanted to perform. However, as there are overwhelming numbers of candidates to be detected by FACS we decided to give up on determining cell types. Instead, we planned to conduct the single cell RNA-seq to estimate the rough cell type repertoire based on their gene expression profiles in the near future, as described in the discussion section.

Reviewer #2: In the manuscript “Whole embryonic detection of maternal microchimeric cells highlights significant differences in their numbers among individuals” the authors introduce a technique to examine the quantity of allogeneic cells in a murine model. Utilizing a breeding strategy to produce GFP- embryos in a GFP+ mother, the authors were able to utilize GFP+ cell presence in the embryos as an indicator of maternal microchimerism. To measure the quantity of GFP+ cells in the embryos they implemented a fluorescence-activated cell sorting strategy to filter cells on PI staining (for live cell detection) and GFP (for allogeneic cell detection). The presence of GFP+ cells, presumably from the mother, were identified in several of the embryos with one presenting with a notably larger proportion of GFP+ cells. This technique provides a useful tool for the analysis of maternal microchimerism in embryo development and may help to answer some of the enigmas of maternal microchimerism. This is an interesting report on a technique that can aid researchers in the understanding of fetomaternal chimerism and mechanisms in utero. I found the methods to be generally well described and took particular interest in the study design. I also have some comments for the authors to consider for further improvements in clarity and dissemination of their findings:

Major comments:

1. The sample size of the study is not clear from the text or figures. Samples sizes listed in lines 196-198 suggest a smaller number of samples compared to figure 3B. It would be helpful to clearly indicate how many litters and embryos were studied at each gestational age for the experiments.

> We thank the reviewer for a constructive comment. To clarify this point, we have added the detailed information of sample sizes both for Figure 2B and Figure 3B, and further provided detailed information as an additional table (Table 1). 

 <Figure 3B legend> l. 239 ~ 242 in ‘Manuscript’

Sample numbers are as follows. control: E12.5: n = 5, E13.5: n = 5, E14.5: n = 5, E15.5: n = 5, E16.5: n = 5, E18.5: n = 4, sample: E12.5: n = 6, E13.5: n = 7, E14.5: n = 4, E15.5: n = 5, E16.5: n = 7, E18.5: n = 4]. See also Table 1 for more detail. 

<Table1 1> l. 246 in ‘Manuscript’

Please refer to the table we have newly added. 

2. The results section should include either a detailed section or table containing descriptive statistics for each gestational age group. As the study focus is in the prevalence and proportion of GFP cells at different gestational ages, I would suggest this to include data on GFP cell proportions, measure of variation, proportion of embryos with detectable GFP cells, sample size, etc.

> We have added a new table (Table 1) to provide the detailed information. 

3. Line 91: The text states that “GFP+ embryos (identified by GFP excitation flashlight while embryos are in amnion) were carefully dissected from their sacrificial mother to avoid cross contamination of fetal and maternal cells.” As I read the rest of the study it refers to the study of the GFP- embryos for the presence of GFP+ cells. Is this a mistake or were the GFP+ embryos removed to avoid contamination of the GFP- embryos? If this is correct, perhaps this could be clarified by including in the schematic of Figure 1C.

> Thank you for pointing it out. Yes. It was actually a mistake. We corrected it.

<Materials and Methods> l. 90 ~ 92 in ‘Manuscript’

To obtain suspensions of dissociated cells from the whole embryo, GFP+GFP- embryos (identified by the absence of fluorescence using GFP excitation flashlight while embryos are in amnion) were carefully dissected…

4. Throughout the manuscript there is a continuous assumption that the source of the GFP+ cells is the mother, however there are other hypotheses described in the literature that could explain the source of microchimerism in utero. Assuming 50% of embryos in the litter are heterozygous (GFP+/-) it is difficult to determine the precise source of the cells. Anastomoses of placental vasculature has been previously demonstrated to produce twin-twin transfusion and could be another potential source of allogenic cells in utero. Were there any other measures used to confirm that the cells are of maternal origin? Is there any information or analyses that can be included about the heterozygous GFP+/- prevalence of each litter?

> We totally agree with the comment that GPF+ cells could also be from GFP+/- siblings in the same litter. Although our experimental design is not capable of discriminating cells from GFP+/-

mother or GFP+/- siblings in the same litter, we have analyzed if the embryos with very high GFP+ cells had higher numbers of GFP+/- siblings. The result suggested that the ratio of GFP+/- siblings may not explain the high frequency of GFP+ cells. We have added the following sentences in the discussion. 

<Discussion> l. 262 ~ 272 in ‘Manuscript’ 

To add, another possibility was that the GFP+ cells identified here could be from GFP+/- siblings in the same liter, however, considering that the litter with the sample had relatively lower ratio of GFP+/- siblings compared to those of the other stages (the ratio of GFP+/- : GFP-/- embryos of the E18.5 was 0.23:1 in the sample litter, where it was 1:1.2 on average in the litters of the other stages. See also Table 1), and we also avoided using embryos with conjugated placenta, the probability of this sibling origin scenario appears to be less likely. Even a minor possibility would be that the embryo was a tetragametic chimera, leading to the high frequency of GFP+ cells, however, at least we could not find any sign of abnormality, including body size, morphology, and patchy signals of GFP. Thus, it would be fair to conclude that most of the cells detected here were MMc cells, or at least immunologically non-self cells.

5. Lines 214-217: Could you elaborate on the possible causes of the seemingly low number of GFP+ cells measured in the embryos? This could perhaps be added as part of a broader discussion of limitations.

> We do not have definite answer to this, however, one possibility would be that the whole process took rather long time, and used FACS machine for soting. We have added this point in the discussion section. 

<Discussion> l. 282 ~ 287 in ‘Manuscript’ 

Finally, major caveats of our methodology would be that 7 – 8 hours are needed to process embryos to obtain single celled suspension, and is not best suited to the mouse embryos from the latest stages. This, together with automatically removed cells in FACS as an electronic aborts (around 1 in 100 counts in our experiments) may explain the reason why we could not detect GFP+ cells in some of the embryos. 

6. Lines 249-255: It is stated that the large proportion of GFP cells identified in one sample is a product of the mother, but it should be considered that this may be a result of tetragametic chimerism (the fusion of two embryos in utero) or possible transfusion from another embryo.

> That is a very important point, thank you. We have added the following sentence to our manuscript.

<Discussion> l. 262 ~ 272 in ‘Manuscript’

To add, another possibility was that the GFP+ cells identified here could be from GFP+/- siblings in the same liter, however, considering that the litter with the sample had relatively lower ratio of GFP+/- siblings compared to those of the other stages (the ratio of GFP+/- : GFP-/- embryos of the E18.5 was 0.23:1 in the sample litter, where it was 1:1.2 on average in the litters of the other stages. See also Table 1), and we also avoided using embryos with conjugated placenta, the probability of this sibling origin scenario appears to be less likely. Even a minor possibility would be that the embryo was a tetragametic chimera, leading to the high frequency of GFP+ cells, however, at least we could not find any sign of abnormality, including body size, morphology, and patchy signals of GFP. Thus, it would be fair to conclude that most of the cells detected here were MMc cells, or at least immunologically non-self cells.

Minor comments

1. Lines 17-19: In the abstract it is stated that “maternal cells…migrate into the fetus and persist for the rest of their lives”, this appears to be an overstatement. While possible, there does not seem to be sufficient evidence to support this claim. It may be best to rephrase this into a phrase similar to “may persist for several decades” (see line 42).

> Thank you. While maternal cells are detected in 89 years old patient with type one diabetes (Chimerism. 2010 Oct-Dec; 1(2): 45–50), we agree to tone down our statement to avoid overstatement. 

<Abstract> l. 17 ~ 19 in ‘Manuscript’

During pregnancy in placental mammals, small numbers of maternal cells (maternal microchimeric cells, or MMc cells) migrate into the fetus and persist decades, or perhaps for the rest of their lives….

2. Line 127: It is not clear if then entire material from each embryo was loaded onto the FACS machine or only a portion.

> Roughly speaking, about one-fourth of total cell suspension volume was loaded onto the FACS machine. We have revised the sentence as follows.

<Materials and Methods> l. 126 ~ 129 in ‘Manuscript’

The cells (> 10,000,000 cells, which roughly corresponds to one-fourth of total cell suspension volume) obtained from a single fetus were stained with Propidium Iodide (PI) solution, and were loaded into a fluorescence-activated cell sorting (FACS) machine (BD AriaⅢu). 

3. In lines 246-249 it is stated that this study identified a comparable number of MMc cells of previous studies reviewed by Kinder et al. However, on review the review article by Kinder et al. does not appear to summarize maternal microchimerism levels (although it does for fetal microchimerism). In this case it would be better to cite the specific studies that produced comparable results.

> Thank you for pointing this out. Actually, while some studies detected MMc cells using FACS in the organs with high MMc frequencies, no other studies analyzed frequency in the whole embryo. To be conservative, we have modified the corresponding part as follows.

<Discussion> l. 254 ~ 256 in ‘Manuscript’

Based on these results, we found that most of the embryos showed a comparable number of MMc cells, around 8 cells / 107 sorted cells (Fig 3B). which is consistent with previous studies reviewed in [4].

---

## [Decision Letter · Decision Letter 1]

13 Oct 2021

PONE-D-21-14310R1Whole embryonic detection of maternal microchimeric cells highlights significant differences in their numbers among individualsPLOS ONE

Dear Dr. Naoki Irie,

Thank you for submitting your manuscript to PLOS ONE. After careful consideration, we feel that it has merit but does not fully meet PLOS ONE’s publication criteria as it currently stands. Therefore, we invite you to submit a revised version of the manuscript that addresses the points raised during the review process.

We thank you for your careful consideration of the reviewers' requests and suggestions, which has greatly improved the manuscript. However, a few clarifications and extra details have been asked by the reviewers, including discussing the influence of embryonic stage on the number of maternal cells.Please submit your revised manuscript by Nov 27 2021 11:59PM. If you will need more time than this to complete your revisions, please reply to this message or contact the journal office at plosone@plos.org. Please include the following items when submitting your revised manuscript:A rebuttal letter that responds to each point raised by the academic editor and reviewer(s). You should upload this letter as a separate file labeled 'Response to Reviewers'.A marked-up copy of your manuscript that highlights changes made to the original version. You should upload this as a separate file labeled 'Revised Manuscript with Track Changes'.An unmarked version of your revised paper without tracked changes. You should upload this as a separate file labeled 'Manuscript'.If applicable, we recommend that you deposit your laboratory protocols in protocols.io to enhance the reproducibility of your results. Protocols.io assigns your protocol its own identifier (DOI) so that it can be cited independently in the future. For instructions see: https://journals.plos.org/plosone/s/submission-guidelines#loc-laboratory-protocols. Additionally, PLOS ONE offers an option for publishing peer-reviewed Lab Protocol articles, which describe protocols hosted on protocols.io. Read more information on sharing protocols at https://plos.org/protocols?utm_medium=editorial-email&utm_source=authorletters&utm_campaign=protocols.

We look forward to receiving your revised manuscript.

Kind regards,

Colette Kanellopoulos-Langevin, Ph.D

Academic Editor

PLOS ONE

Journal Requirements:

Additional Editor Comments (if provided):

Reviewers' comments:

Reviewer's Responses to Questions

**Comments to the Author**

1. If the authors have adequately addressed your comments raised in a previous round of review and you feel that this manuscript is now acceptable for publication, you may indicate that here to bypass the “Comments to the Author” section, enter your conflict of interest statement in the “Confidential to Editor” section, and submit your "Accept" recommendation.

Reviewer #1: All comments have been addressed

Reviewer #2: (No Response)

2. Is the manuscript technically sound, and do the data support the conclusions?

Reviewer #1: Yes

Reviewer #2: Partly

3. Has the statistical analysis been performed appropriately and rigorously? 

Reviewer #1: Yes

Reviewer #2: N/A

4. Have the authors made all data underlying the findings in their manuscript fully available?

Reviewer #1: Yes

Reviewer #2: Yes

5. Is the manuscript presented in an intelligible fashion and written in standard English?

Reviewer #1: Yes

Reviewer #2: Yes

6. Review Comments to the Author

Reviewer #1: please find details in the attached file

The authors have carefully taken into account the comments of the two reviewers.

Table 1 was absolutely essential and allows a substantiated argumentation of data as to the probable maternal origin of these cells.

Moreover this table allows to note that when embryonic stages increase, “maternal” Mc is more abundant. This could be one of the reason why the embryo for which the number of "maternal" cells was the highest was the one tested at the latest embryonic stage.

Thus, although it is undeniable that there is an heterogeneity between individuals for the number of maternal microchimeric cells, since the embryos do not have the same number of Mc cells for the same embryonic stage; it seems there is also an effect due to the embryonic stage.

Indeed a Mann Whitney test allows to see that the number of maternal cells is statistically higher at E16.5 (p=0.0147) compared to all other previous embryonic stages (E12.5 + E13.5 +E14.5 + E15.5).

Thus it is very possible that later embryonic stages are those where MMc is the most perfused. Nevertheless a Spearman correlation does not show a statistical increase with embryonic stages (p=0.3), only a tendency.

The fact that there appears to be an effect due to the embryonic stage needs to be mentioned at least in the discussion and probably in the results.

Reviewer #2: The manuscript “Whole embryonic detection of maternal microchimeric cells highlights significant differences in their numbers among individuals” details a technique for whole embryo analysis of maternal microchimerism. In a murine model the authors were able to utilize a breeding strategy of female mice heterozygous for fluorescent GFP with wild-type males to produce GFP- offspring. Using fluorescence-activated cell sorting, the authors were able to identify the presence of live GFP+ cells in some GFP- embryos, presumably originating from the mother. The study in several embryos found differences in GFP presence and quantity with one embryo presenting with a significantly larger proportion of GFP+ cells. The technique presented is well described and provides a useful approach for investigating the phenomenon of maternal microchimerism. The authors have appropriately addressed the reviewers’ comments and I am satisfied with the revisions made to address these concerns. In addition, I have a few additional comments that the authors should consider for the manuscript:

1. The primary objective stated in line 67 indicates testing the frequency of maternal microchimerism among individual embryos during normal pregnancy. The authors make the conclusion that there is similarity in the quantity of GFP+ cells between the embryos however there appears to be notable variation in qualitative GFP+ cell detection. Now seeing the detail and distribution of results provided in Table 1, there should be some additional discussion regarding variation between litters. For example, litter D and litter H with 100% prevalence of GFP+ cell detection compared to other litters C and I at the same gestational ages.

2. In the discussion section, lines 254-255, it is mentioned that “most of the embryos showed a comparable number of MMc cells, around 8 cells/ 10^7 sorted cells”. However, this estimation contradicts the data presented in Table 1 which (excluding the outlier) average 3.7 cells/10^7 sorted cells among samples with detectable GFP+ cells. Also, to improve clarity it would be helpful to indicate discussion of microchimerism results relating to “all embryos” or “GFP+ cell detected embryos”.

3. Further, in line 256-258 it is described that “one of the embryos at the latest stage (E18.5) showed 1,816 cells/10^7 cells, which corresponds to a frequency 1,000 times higher than that in the detected embryos”. This is true for some embryos with the lowest 1 cell/10^7 cells, but based on the average this would be approximately 500 times higher. Additional detail should be included in this sentence to help clarify the conclusions. This is also mentioned in the abstract line 27, and introduction line 71.

7. PLOS authors have the option to publish the peer review history of their article (what does this mean?). If published, this will include your full peer review and any attached files.

Reviewer #1: **Yes: **Nathalie C. Lambert

Reviewer #2: No

---

## [Author Response · Author response to Decision Letter 1]

4 Nov 2021

Review Comments to the Author

Reviewer #1: please find details in the attached file

The authors have carefully taken into account the comments of the two reviewers. 

Table 1 was absolutely essential and allows a substantiated argumentation of data as to the probable maternal origin of these cells.

Moreover this table allows to note that when embryonic stages increase, “maternal” Mc is more abundant. This could be one of the reason why the embryo for which the number of "maternal" cells was the highest was the one tested at the latest embryonic stage. 

Thus, although it is undeniable that there is an heterogeneity between individuals for the number of maternal microchimeric cells, since the embryos do not have the same number of Mc cells for the same embryonic stage; it seems there is also an effect due to the embryonic stage.

Indeed a Mann Whitney test allows to see that the number of maternal cells is statistically higher at E16.5 compared to all other previous embryonic stages (E12.5 + E13.5 +E14.5 + E15.5). 

Thus it is very possible that later embryonic stages are those where MMc is the most perfused. Nevertheless a Spearman correlation does not show a statistical increase with embryonic stages (p=0.3), only a tendency. 

The fact that there appears to be an effect due to the embryonic stage needs to be mentioned at least in the discussion and probably in the results. 

> We sincerely appreciate your detailed investigation by further re-analyzing our raw data. Actually, we have performed similar analyses, but were a bit too conservative to refer in our manuscript. We have added this information to our manuscript. 

 Discussion Section l.272

Finally, considering that later stages are known to show higher frequency of MMc cells [23], it is intriguing that the sample with highly frequent MMc cells was detected for the latest embryonic stage. Consistently, while no statistically significant correlation was obtained between the number of MMc cells and developmental stages, however, we still found a weak tendency even without stage 18.5 (Spearman’s correlation coefficiency 0.3 for stages from 12.5 to 16.5).

Reviewer #2: The manuscript “Whole embryonic detection of maternal microchimeric cells highlights significant differences in their numbers among individuals” details a technique for whole embryo analysis of maternal microchimerism. In a murine model the authors were able to utilize a breeding strategy of female mice heterozygous for fluorescent GFP with wild-type males to produce GFP- offspring. Using fluorescence-activated cell sorting, the authors were able to identify the presence of live GFP+ cells in some GFP- embryos, presumably originating from the mother. The study in several embryos found differences in GFP presence and quantity with one embryo presenting with a significantly larger proportion of GFP+ cells. The technique presented is well described and provides a useful approach for investigating the phenomenon of maternal microchimerism. The authors have appropriately addressed the reviewers’ comments and I am satisfied with the revisions made to address these concerns. In addition, I have a few additional comments that the authors should consider for the manuscript:

> We sincerely appreciate your encouraging comments.

1. The primary objective stated in line 67 indicates testing the frequency of maternal microchimerism among individual embryos during normal pregnancy. The authors make the conclusion that there is similarity in the quantity of GFP+ cells between the embryos however there appears to be notable variation in qualitative GFP+ cell detection. Now seeing the detail and distribution of results provided in Table 1, there should be some additional discussion regarding variation between litters. For example, litter D and litter H with 100% prevalence of GFP+ cell detection compared to other litters C and I at the same gestational ages.

> This is actually a novel, important point that we overlooked. Thank you for the inspiring comment, and we have added the following sentence in our discussion part.

Discussion Section l.283

With this respect, it is tempting to know if the ratio of MMc+ detected embryos and those undetected differ among the litters (e.g., at least Litter C vs D, Litter H showed statistically significant ratio, see also Table 1).

2. In the discussion section, lines 254-255, it is mentioned that “most of the embryos showed a comparable number of MMc cells, around 8 cells/ 10^7 sorted cells”. However, this estimation contradicts the data presented in Table 1 which (excluding the outlier) average 3.7 cells/10^7 sorted cells among samples with detectable GFP+ cells. Also, to improve clarity it would be helpful to indicate discussion of microchimerism results relating to “all embryos” or “GFP+ cell detected embryos”.

> We totally agree with the comment, and we have made following modification to our manuscript. 

Discussion Section l.254

Based on these results, we found that majority of the embryos showed a comparable number of MMc cells. Meanwhile, an unexpected finding was that one of the embryos at the latest stage (E18.5) showed 1,816 cells/107 cells, which corresponds to a frequency approximately 500 times higher than the average of the other detected embryos (around 3.7 cells/ 107 sorted cells for those detected with GFP+ cells but without the one with 1816 cells. See Fig 3B and Table 1).

3. Further, in line 256-258 it is described that “one of the embryos at the latest stage (E18.5) showed 1,816 cells/10^7 cells, which corresponds to a frequency 1,000 times higher than that in the detected embryos”. This is true for some embryos with the lowest 1 cell/10^7 cells, but based on the average this would be approximately 500 times higher. Additional detail should be included in this sentence to help clarify the conclusions. This is also mentioned in the abstract line 27, and introduction line 71.

> We have modified as follows. 

Discussion Section l.254

Meanwhile, an unexpected finding was that one of the embryos at the latest stage (E18.5) showed 1,816 cells/107 cells, which corresponds to a frequency approximately 500 times higher than that in the detected embryos.

In addition, we also corrected the corresponding parts in the abstract and introduction.

Abstract l.26

Using this technique, we found that the number of MMc cells was comparable in most of the analyzed embryos; however, around 500 times higher number of MMc cells was detected in one embryo at the latest stage.

Introduction l.69

We succeeded in developing an effective method for detecting whole embryonic MMc cells and found that the number of MMc cells was comparable in most of the analyzed embryos; however, around 500 times higher number of MMc cells was detected in one embryo at the latest stage.

---

## [Decision Letter · Decision Letter 2]

1 Dec 2021

Whole embryonic detection of maternal microchimeric cells highlights significant differences in their numbers among individuals

PONE-D-21-14310R2

Dear Dr. Naoki Irie,

We’re pleased to inform you that your manuscript has been judged scientifically suitable for publication and will be formally accepted for publication once it meets all outstanding technical requirements.

One minor requirement has been made by one reviewer regarding the most recent revision, and I agree with his suggestion:

 1. Lines 283-285: The added sentence regarding my previous comment on examining the ratio of GFP+ and GFP- embryos does not read clearly and may need to be revised**.** I believe the statement in parentheses is trying to say Litter C vs D and Litter H vs I show significant variation in the litter chimerism ratio at the same gestational age. Perhaps more concisely, **“the litters at stages E13.5 and E16.5 show significant variation in the proportion of MMc+ detected embryos”.**

Kind regards,

Colette Kanellopoulos-Langevin, Ph.D

Academic Editor

PLOS ONE

Additional Editor Comments (optional):

Reviewers' comments:

Reviewer's Responses to Questions

**Comments to the Author**

1. If the authors have adequately addressed your comments raised in a previous round of review and you feel that this manuscript is now acceptable for publication, you may indicate that here to bypass the “Comments to the Author” section, enter your conflict of interest statement in the “Confidential to Editor” section, and submit your "Accept" recommendation.

Reviewer #1: All comments have been addressed

Reviewer #2: All comments have been addressed

2. Is the manuscript technically sound, and do the data support the conclusions?

Reviewer #1: Yes

Reviewer #2: Yes

3. Has the statistical analysis been performed appropriately and rigorously? 

Reviewer #1: Yes

Reviewer #2: Yes

4. Have the authors made all data underlying the findings in their manuscript fully available?

Reviewer #1: Yes

Reviewer #2: Yes

5. Is the manuscript presented in an intelligible fashion and written in standard English?

Reviewer #1: Yes

Reviewer #2: Yes

6. Review Comments to the Author

Reviewer #1: All comments have been adequately addressed by the authors in this second revision. In particular

the fact that it is very possible that later embryonic stages are those where MMc is the most perfused, although statistical analyses show only a trend toward a correlation.

Reviewer #2: The authors of the manuscript “Whole embryonic detection of maternal microchimeric cells highlights significant differences in their numbers among individuals” describe their research using a technique to examine whole embryo microchimerism in murine models. This technique provides a novel insight into early stage transplacental maternal microchimerism and additional discoveries of variation in microchimerism presentation with the occasional incidence of significantly large cell populations. The authors have addressed the comments from both reviewers and made appropriate revisions to the manuscript. The work has shown significant improvement with the additional data and analyses that the authors have subsequently integrated into the final manuscript. I only have a minor comment regarding the most recent revisions.

1. Lines 283-285: The added sentence regarding my previous comment on examining the ratio of GFP+ and GFP- embryos does not read clearly and may need to be revised. I believe the statement in parentheses is trying to say Litter C vs D and Litter H vs I show significant variation in the litter chimerism ratio at the same gestational age. Perhaps more concisely, “the litters at stages E13.5 and E16.5 show significant variation in the proportion of MMc+ detected embryos”.

7. PLOS authors have the option to publish the peer review history of their article (what does this mean?). If published, this will include your full peer review and any attached files.

Reviewer #1: No

Reviewer #2: **Yes: **Brandon N. Johnson

---

## [Editor Report · Acceptance letter]

13 Dec 2021

PONE-D-21-14310R2 

Whole embryonic detection of maternal microchimeric cells highlights significant differences in their numbers among individuals 

Dear Dr. Irie:

I'm pleased to inform you that your manuscript has been deemed suitable for publication in PLOS ONE. Congratulations! Your manuscript is now with our production department. 

Kind regards, 

on behalf of

Dr. Colette Kanellopoulos-Langevin 

Academic Editor

PLOS ONE